# Synthesis and Characterization of New Ruthenium (II) Complexes of Stoichiometry [Ru(*p*-Cymene)Cl_2_L] and Their Cytotoxicity against HeLa-Type Cancer Cells

**DOI:** 10.3390/molecules27217264

**Published:** 2022-10-26

**Authors:** Marta G. Fuster, Imane Moulefera, Mercedes G. Montalbán, José Pérez, Gloria Víllora, Gabriel García

**Affiliations:** 1Departamento de Ingeniería Química, Facultad de Química, Campus Regional de Excelencia “Campus Mare Nostrum”, Universidad de Murcia, 30071 Murcia, Spain; 2Departamento de Ingeniería Química y Medioambiental, ETSII, Universidad Politécnica de Cartagena, 30203 Cartagena, Spain; 3Departamento de Química Inorgánica, Facultad de Química, Campus Regional de Excelencia “Campus Mare Nostrum”, Universidad de Murcia, 30071 Murcia, Spain

**Keywords:** ruthenium, complexes, cytotoxicity, anticancer activity, HeLa, MTT assay

## Abstract

When the [Ru(*p*-cymene)(μ-Cl)Cl]_2_ complex is made to react, in dichloromethane, with the following ligands: 2-aminobenzonitrile (2abn), 4-aminobenzonitrile (4abn), 2-aminopyridine (2ampy) and 4-aminopyridine (4ampy), after addition of hexane, the following compounds are obtained: [Ru(*p*-cymene)Cl_2_(2abn)] **(I)**, [Ru(*p*-cymene)Cl_2_(4abn)] **(II)**, [Ru(*p*-cymene)Cl_2_(2ampy] **(III)** and [Ru(*p*-cymene)Cl_2_(μ-(4ampy)] **(IV)**. All the compounds are characterized by elemental analysis of carbon, hydrogen and nitrogen, proton nuclear magnetic resonance, COSY ^1^H-^1^H, high-resolution mass spectrometry (ESI), thermogravimetry and single-crystal X-ray diffraction (the crystal structure of **III** is reported and compared with the closely related literature of **II**). The cytotoxicity effects of complexes were described for cervical cancer HeLa cells via 3-(4.5-dimethylthiazol-2-yl)-2.5-diphenyltetrazolium bromide (MTT) assay. The results demonstrate a low in vitro anticancer potential of the complexes.

## 1. Introduction

In addition to *cis*-platinum [1] and other similar compounds such as carbo-platin [2,3,4] or allyl-platinum [5,6,7], it has also been shown that numerous ruthenium (II) compounds are able to exhibit similar properties for cancer therapy [8]. Figure 1 presents some of those that lately have received special attention [9,10,11,12,13].

Recently, the use of platinum-based compounds has been limited because of their lack of cellular selectivity and drug resistance. Ruthenium is attracting the interest of researchers as a promising alternative to platinum-based complexes due to their several oxidation states, Ru (II), Ru (III) and Ru (IV), and improved cellular selectivity [14]. In this way, ruthenium (II) complexes display excellent structural, photophysical and biological properties, which makes them promising anticancer compounds [15]. The mechanism of action for many ruthenium complexes differs from the DNA-binding mechanism typically associated with platinum compounds, having a wider range of intracellular targets [16]. In addition, ruthenium-based drugs have demonstrated superior therapeutic efficiency in robust metastatic cancers or cisplatin resistant tumors [17].

Some years ago, our research group described the synthesis of the compound [Ru(*p*-cymene)Cl(o-phen)]PF_6_, (o-phen = o-phenylenediamine) [18], which has shown to be an excellent antitumor agent [19,20]. Moreover, complexes [Ru(*p*-cymene)Cl_2_L], L being one of the following substituted pyridines: 2-fluoro-5-aminopyridine, 5-amino-2-chloropyridine and 2-bromo-5-aminopyridine, have been isolated and characterized and have shown moderate cytotoxicity against lung carcinoma A549 and breast cancer MCF-7 cell lines [21]. Recently, J. Ruiz et al. [22] have described a set of ruthenium (II) complexes containing *p*-cymene, and a C-N ligand with a non-coordinated CHO group, capable of exhibiting anticancer properties towards several human cancer cell lines, including cells of the epithelial ovarian carcinoma A2780, CDDP-resistant ovarian cancer A2780cisR and breast cancer MCF-7. They have also tested the compounds in the non-tumorigenic BGM and CHO cells, finding high selectivity towards cancer cells over normal cells. Ismail et al. [23] have synthesized two new hybrid half-sandwiched Ru(II) arene complexes of a general formula [*η*^6^-(*p*-cymene)Ru(L)Cl] (where L = *1-(Benzazol-2-yl)-3-(thiophen-2-yl) propane-1,3-dione*), which have exhibited significant inhibitory activity against human breast and lung cancer cells (MCF-7 and A549), finding lower IC_50_ values than those of clinical cisplatin drugs. Recently, our group has prepared new compounds of ruthenium (II), neutral and ionic that behave as anticancer agents, especially one of them: [Ru(*p*-cymene)(2amfol)Cl_2_]_;_, (2amfol = 2-aminephenol) against HeLa and MCF-7 cells [24].

Nowadays, scientists worldwide have expanded the use of specific features of organometallic ruthenium compounds (e.g., structural diversity, ligand exchangeability, redox and catalytic properties) for medicinal purposes with surprising results. The aim of this work is the synthesis and characterization of organometallic ruthenium compounds, obtained by reacting [Ru(*p*-cymene) Cl(μ-Cl)]_2_ against potentially bidentate ligands with two donor nitrogen atoms, obtaining the new complexes [Ru(*p*-cymene) Cl_2_L]; L = 2abn, 4abn, 2ampy and 4ampy. In addition, a study of their properties as antitumor agents was carried out by MTT assay on HeLa cells.

## 2. Results and Discussion

Although the ligands used are potentially bidentate, they always act as monodentate across the amine nitrogen (abn) or pyridine nitrogen (ampy). This behavior has been previously observed on ruthenium (II) complexes [18,24]. Only D. S. Pandey et al. have described a related compound in which the CN group acts as a bridge [25].

The four ruthenium (II) complexes shown in Figure 2 were obtained with good yields, and they are air-stable solids that present colors in the yellow–orange range. All of them gave satisfactory analyses of carbon, hydrogen and nitrogen (see Table 1). Moreover, all the isolated complexes are neutral in acetone solutions 5 × 10^−4^ M [25].

The X-ray diffraction structures of complexes **II** [26] and **III** [27] have been previously described, however, in the case of **II,** no additional data have been presented and, in the case of **III**, in our hands, another polymorph with a different space group is obtained.

### 2.1. Molecular Structure

Complexes exhibit distorted octahedral geometry: three positions occupied by chlorine and N atoms and the three remaining by *p*-cymene ligand. The highest deviation from the ideal 90 degree bond angle is N-RuCl2, 80.17^o^ (**II**) and N-Ru-Cl2, 86.22° (**III**). Most relevant molecular parameters are shown in Table 2. The difference in the hybridization of coordinated nitrogen explains the difference in Ru-N distances: coordinated N in **II** is sp3 and in **III** is sp (see Figure 3). Both Cl-Ru distances are similar in RuCl2(*p*-cymene) (4-aminobenzonitrile), but in RhCl_2_(*p*-cymene) (2-aminopyridine), Cl-Rh distances are different (Table 2); the reason is an intramolecular hydrogen bond N-H···Cl that lengthens Rh-Cl(1) distance.

Difference in molecule surface and volume (Table 2) matches differences in these parameters for ligands 4-aminobenzonitrile and 2-aminopyridine. These ligands also originally differ in ovality.

### 2.2. Supramolecular Structure

The most relevant features of crystal packing in **II** are shown in Figure 4. The structure consists of infinite chains of molecules linked by N-H···N≡C contacts. In addition, there are C-H···Cl and N-H···Cl contacts. This is confirmed by the Hirshfeld surface (HS) analysis [28,29,30,31]. Figure 5 shows the d_norm_ Hirshfeld surface; its associated 2D fingerprint can be decomposed into contributions from N···H/H···N interactions (these make up 13.8% of the surface area of the HS) and H···Cl/Cl···H interactions (21.1%). The Appendix A contains fingerprint plots.

In spite of complex **III** possessing smaller molecular weight, it shows larger density than **II**. This more compact crystal packing for **III** can be explained in part by the π···π stacking interaction between 2ampy ligands (Figure 4) with centroids at 3.695 Å distance. In addition, there are C-H···π and N-H···Cl contacts. Figure 5 shows the d_norm_ Hirshfeld surface; its associated 2D fingerprint can be decomposed into contributions from C···H/H···C interactions (these make up 10.5% of the surface area of the HS), H···Cl/Cl···H interactions (21.8%) and C···C interactions (3.9%). The Appendix A contains fingerprint plots.

Complexes **I** and **II** show absorptions due to ν(CN) in the 2230 cm^−1^ environment that agree that the ligand does not behave as a bridge through nitrile nitrogen. However, the three bands assignable to NH_2_ appear between 3200 and 3100 cm^−1^. The coordination through the pyridine nitrogen atom in **III** and **IV** is evidenced by the displacement of two bands at 604 cm^−1^ (in-plane ring deformation) and 405 cm^−1^ (out-of-plane ring deformation) to higher frequencies (631 and 424 cm^−1^, respectively). Complexes **I**, **II**, **III** and **IV** exhibit bands assignable to ν(NH) in the 3400–3100 range, about 100 cm^−1^ below the free ligand [32]. In **I** and **II**, the decrease in NH_2_ frequencies is due to the coordination of amine group to the ruthenium atom. However, in the case of **III** and **IV**, where the neutral ligand is bound to the ruthenium via the pyridinic nitrogen, the decrease in NH_2_ frequencies may be due to intra-(**III**) or inter-(**IV**) molecular NH_2_-Cl interactions. All the isolated compounds show two absorptions assigned to ν(RuCl) at about 290–270 cm^−1^.

Table 1 also includes the data obtained from their high-resolution mass spectra (ESI), and Table 3 show relevant data from Proton Nuclear Magnetic Resonance. The assignments of the aromatic ring signals of the ligands (2abn and 2ampy) in **I** and **III** were made through COSY ^1^H-^1^H experiments.

In relation to the mass spectra, under the working conditions used, the molecular peak corresponding to the exact mass is not observed, but others due to [M-Cl]^+^ are observed. This type of behavior has been previously observed [22,24].

### 2.3. In Vitro Cytotoxicity

The in vitro cytotoxic effects of Complex **I**, Complex **II**, Complex **III** and Complex **IV** were assessed with the MTT assay after 48 h of exposure in the HeLa cell line. This cell line was selected due to its origin to determine if any of the complexes have anticancer activity. HeLa cells are derived from a cervical carcinoma and have been widely used in cytotoxicity studies [33]. To study the effects of the complexes, concentrations between 0.031–2 mM were evaluated. As complexes were dissolved in DMSO (0.5% *v*/*v*), this solvent without complex was used at the same concentration as the control. Cells without treatment were used as the control (data not revealed). After 48 h of incubation with several concentrations of the complexes, the cell viability decreased very slowly or remained almost constant as the concentration increased (Figure 6). However, Complex IV was more cytotoxic than the others, revealing an IC_50_ value, defined as the minimal concentration of a drug that is required for 50% inhibition in vitro of 1.6 ± 0.004 mM.

Table 4 shows the cytotoxicity expressed as IC_50_ mean values (mM) of the complexes synthesized and cisplatin exposed to HeLa cells for 48 h. As can be seen, the cytotoxic potency is negligible on the HeLa cell line studied.

## 3. Materials and Methods

The solvents were dried by conventional methods. The ligands were commercial-grade chemicals and [Ru(*p*-cymene)Cl_2_]_2_ were prepared by the method published in [34]. ^1^H NMR spectra were recorder on a Bruker Advance 200, 300 or 400 MHz instrument. IR spectra were recorded on a 100 FT-IR Spectrometer as Nujol mulls. The C, H and N analyses were obtained with a LECO CNHS-932 elemental microanalyzer. Thermal decomposition studies were carried out on a simultaneous analyzer TGA-DTA of TA Instruments. High-resolution (HR)-ESI-MS spectrometry was realized by HPLC Spectrometer/MS TOF Agilent Model 6220.

### 3.1. X-ray Data Collection, Structure Solution and Refinement for II and III

Diffraction data were collected on a Bruker D8 QUEST diffractometer with monochromated Mo-Kα radiation (Kα = 0.71073 Å), performing ω and φ scans at 100(2) K. SADABS 2016/2 absorption correction was performed (Bruker (2016). *APEX3*, *SADABS* and *SAINT*. Bruker AXS Inc., Madison, Wisconsin, USA)

The structures were solved by direct methods [35] and refined anisotropically on F2 using the program SHELXL-2018 [36]. CIF files were validated using checkCIF [37]. Numerical details are presented in Table 5. Hydrogen atoms bound to C were located at geometrically idealized positions and were allowed to ride on the parent atoms; hydrogen atoms bound to N atoms were discernible from difference-Fourier maps and were subsequently refined with N-H distance restraints. Crystal data for **II** were reported previously [26]; we collected data at low temperature in order to achieve a more precise structural study.

The generated Hirshfeld surfaces and the associated 2D fingerprint plots are extracted using the CrystalExplorer17 software [38].

### 3.2. Synthesis of the Complexes

These were prepared according to the following procedure, very similarly to reference [18]. To a dichloromethane (15 mL) solution of [(*p*-cymene)RuCl_2_]_2_ (0.4902 mmol), the appropriate ligand (0.9804 mmol for **I**, **II** and **III**; and 0.4902 mmol for **IV**) was added. The resulting suspension was stirred for 1 h and then was concentrated. The solid obtained was separated by filtration, crystallized from dichloromethane/ether and repeatedly washed with diethyl ether.

The crystals used for X-ray diffraction were obtained by slow diffusion of hexane over solutions of **II** and **III** in dichloromethane.

### 3.3. Cytotoxicity Assays

#### 3.3.1. Cell Culture

Human cervical cancer cells (HeLa) were acquired from the American Type Tissue Culture Collection (ATCC, USA). Cell lines were maintained in the Dulbecco’s Modified Eagle Medium (DMEM) with a low content of glucose (1 g/L) supplemented with 10% (*v*/*v*) fetal bovine serum (FBS), 1 mM glutamax, 1% antibiotics (penicillin-streptomycin) and 1 mM pyruvate. Cells were subcultured, and medium was changed once a week. A total of 0.25% trypsine-0.25 mM EDTA was used.

#### 3.3.2. MTT Assay

A total of 3000 cells/well of HeLa carcinoma cell line were seeded onto 96-well plates and incubated at 37 °C in 5% CO_2_. After 24 h, the culture medium of each well was replaced with fresh medium, and cells were treated with different concentrations of Complex **I**, Complex **II**, Complex **III** and Complex **IV**; these solutions were diluted in DMSO to obtain a maximum concentration per well of 2 mM. In each experiment, growth medium without nanoparticles was used as a control. Cells were incubated at 37 °C for 48 h. Then, the media were removed and 200 µL of 1% MTT (3-(4,5-dimethylthiazol-2-yl)-2,5-diphenyltetrazolium bromide) was added. After 4 h of incubation, the MTT was removed and 100 µL of dimethyl sulfoxide DMSO was added. The absorbance was measured at 560 nm in a microplate reader Fluostar Omega spectrophotometer. Each sample was tested in three independent sets with triplicate points.

## 4. Conclusions

Four ruthenium(II) complexes were designed and synthesized, namely [Ru(*p*-cymene)Cl_2_(2abn)] **(I)**, [Ru(*p*-cymene)Cl_2_(4abn)] **(II)**, [Ru(*p*-cymene)Cl_2_(2ampy] **(III)** and [Ru(*p*-cymene)Cl_2_(μ-(4ampy)] **(IV)**. Good yields were obtained, and all are air-stable solids exhibited colors in the yellow–orange range. All the compounds were fully characterized by elemental analysis of carbon, hydrogen and nitrogen, proton nuclear magnetic resonance, COSY ^1^H-^1^H, high-resolution mass spectrometry (ESI), thermogravimetry and single-crystal X-ray diffraction. Furthermore, the cytotoxic effect of the complexes was evaluated against HeLa cells, and cell viability was found to decrease very slowly in all cases. Only complex IV was more cytotoxic than the others, but it presented a much higher IC_50_ than cisplatin. However, although the results of the synthesized complexes do not show high cytotoxicity values against HeLa cells compared with cisplatin, further studies on the synthesis and characterization of ruthenium complexes are certainly very useful to be able to relate the structures to their unique and versatile biochemical properties.

## Figures and Tables

**Figure 1 molecules-27-07264-f001:**
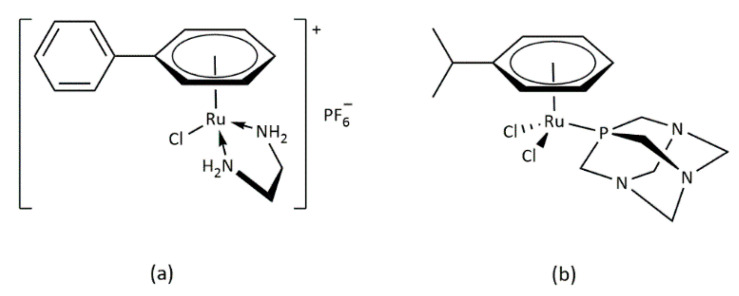
Structures of some antitumor active ruthenium complexes. (**a**) [(*η^6^*-biphenil)Ru(en)Cl]PF_6_ and (**b**) [(*η^6^*-p-cymene)Ru(pta)Cl_2_].

**Figure 2 molecules-27-07264-f002:**
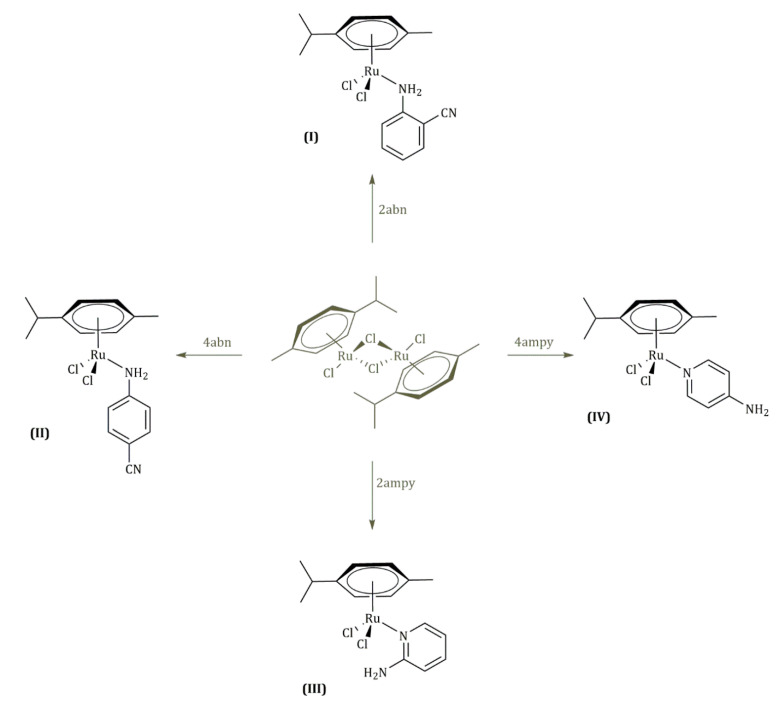
Complexes **I**, **II**, **III** and **IV**.

**Figure 3 molecules-27-07264-f003:**
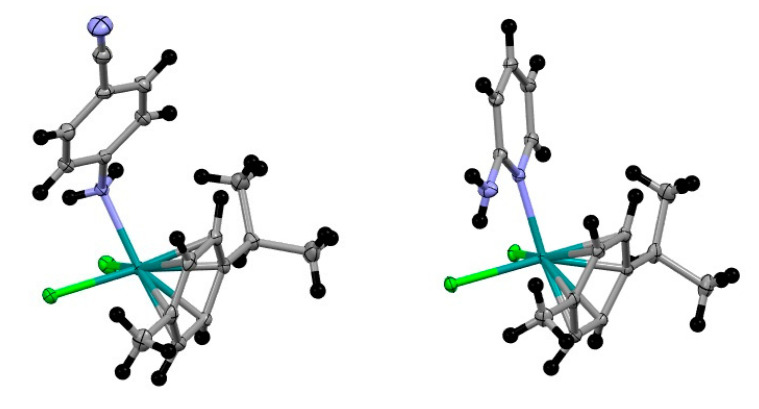
Molecular structure of **II** (**left**) and **III** (**right**).

**Figure 4 molecules-27-07264-f004:**
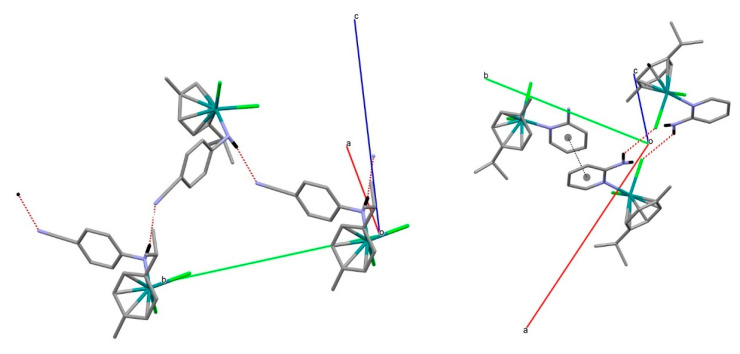
Supramolecular packing in **II** (**left**) and **III** (**right**).

**Figure 5 molecules-27-07264-f005:**
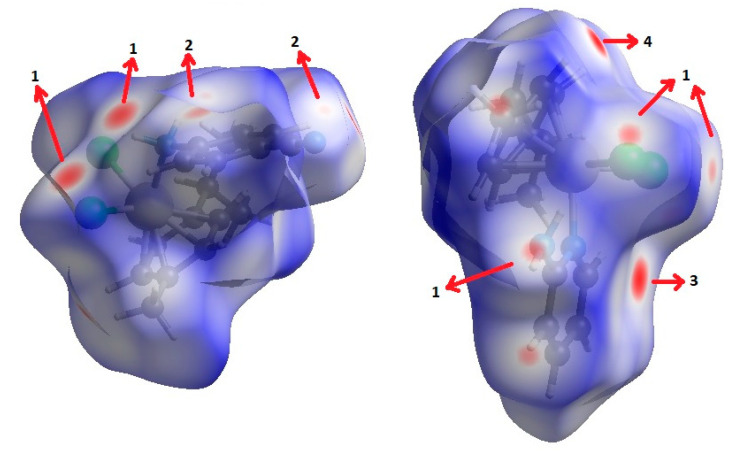
View of the Hirshfeld surfaces mapped over d_norm_ property of **II** (**left**) and **III** (**right**). The labels 1, 2, 3 and 4 represent Cl···H/H···Cl, N···H/H···N, π···π and C-H···π interactions, respectively.

**Figure 6 molecules-27-07264-f006:**
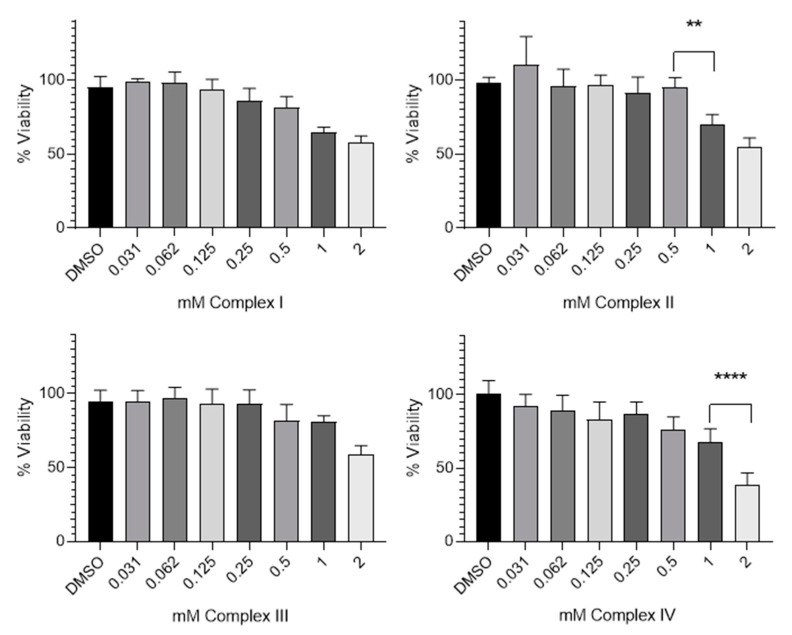
Cytotoxicity effect of Complex **I**, Complex **II**, Complex **III** and Complex **IV** on HeLa cell line. Data are expressed as percentage of cell viability ± SD versus concentration. ** indicates *p* < 0.01 and **** indicates *p* < 0.0001, compared with the adjacent column.

**Table 1 molecules-27-07264-t001:** Color, yield, Elemental Analysis, Exact Mass and Decomposition temperatures from complexes **I**, **II**, **III** and **IV**.

Comp.	Color	Yield (%)	Analysis ^a^ (%)			Exact Mass ^a^ (g/mol)			M.P. ^b^ (°C)
			C	H	N	[M-H]^+^	[M-Cl]^+^	[M-2Cl]^+^	
**I**	Yellow–orange	70	48.09 (48.12)	4.73 (4.75)	6.40 (6.60)		388.8792 (388.8812)	-	69
**II**	Yellow–orange	90	48.25 (48.12)	4.77 (4.75)	6.53 (6.60)	425.3591 (425.3418)	-	-	90
**III**	Orange	85	44.86 (45.01)	5.06 (5.04)	6.83 (7.00)	-	-	329.0595 (329.4365)	78
**IV**	Light Brown	90	44.70 (45.01)	4.87 (5.04)	6.95 (7.00)	-		329.0599 (329.4365)	72

^a^ Calculates values in parenthesis. ^b^ Decomposition temperatures from the thermogravimetric curves.

**Table 2 molecules-27-07264-t002:** Relevant molecular parameters for **II** and **III**.

RuCl_2_(*p*-Cymene)(4-Aminobenzonitrile)		RhCl_2_(*p*-Cymene)(2-Aminopyridine)	
**Ru-N**	2.1756(18) Å	Ru-N	2.1693(14) Å
**Ru-Cl(1)**	2.4260(5) Å	Ru-Cl(1)	2.4545(8) Å
**Ru-Cl(2)**	2.4219(5) Å	Ru-Cl(2)	2.4153(8) Å
**N-Ru-Cl(1)**	82.57(5)^o^	N-Ru-Cl(1)	88.14(3)^o^
**N-Ru-Cl(2)**	80.17(5)^o^	N-Ru-Cl(2)	88.86(3)^o^
**Cl(1)-Ru-Cl(2)**	89.261(17)^o^	Cl(1)-Ru-Cl(2)	86.22(3)^o^
**Ru-Centroide(*p*-cymene)**	1.421 Å	Ru-Centroide(*p*-cymene)	1.432 Å
		Hydrogen bond N(2)-H(01)···Cl(1) [Å and ^o^] d(N-H): 0.851(16); d(H···Cl): 2.418(17); d(N···Cl): 3.1917(16); <(NHCl): 151(2).	
**Molecular surface**	537.2 Å^2^	Molecular surface	491.6 Å^2^
**Molecular volume**	299.9 Å^3^	Molecular volume	280.8 Å^3^
**Ovality**	1.730	Ovality	1.669

**Table 3 molecules-27-07264-t003:** ^1^H-NMR data of **I**, **II**, **III** and **IV**.

Complex	^1^H δ(SiMe_4_) (in CD_2_CCl_2_)	Ligand Structure
**I**	7.37–7.28 (m, **H****b** + **H****d**) 6.82 (d, **H****a**, Jab = 8.1Hz) 6.67 (t, **H****c**, Jbc = 8.1Hz) 5.88–5.72 (dd, –C_6_**H_4_**, J = 6.2 Hz) 5.47 (s, *br*, –N**H_2_**) 2.78 (spt, 1H, –C**H**(CH_3_)_2_, J = 6.9 Hz) 1.61 (s, –C**H_3_**) 1.30 (d, –CH(C**H_3_**)**_2_**, J = 6.9 Hz)	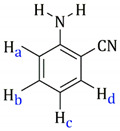
**II**	7.23 (m, **H****a**) 6.54 (m, **H****b**) 5.75–5.53 (dd, –C_6_**H_4_**, J = 6.2 Hz) 4,72 (s, *br*, –N**H_2_**) 2.65 (spt, –C**H**(CH_3_)_2_) 2.07 (s, –C**H_3_**) 1.18 (d, 6H, –CH(C**H_3_**)**_2_**, J = 6.8 Hz)	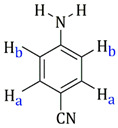
**III**	8.54 (d, **H****d**, J = 4.8 Hz) 7.42 (pst, **H****c**) 6.62 (pst, **H****b**) 6.58 (d, **H****a**, J = 8.0 Hz) 6.12 (s, *br*, –N**H_2_**) 5.53–5.31 (dd, –C_6_**H_4_**, J = 6.0 Hz) 2.93 (spt, –C**H**(CH_3_)_2_, J = 7.2 Hz) 1.96 (s, –C**H_3_**) 1.28 (d, 6H, –CH(C**H_3_**)**_2_**, J = 6.8 Hz)	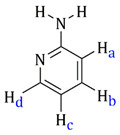
**IV**	8.33 (d, **H****a**, J = 6.9 Hz) 6.43 (d, **H****b**, J = 6.9 Hz) 4.55 (s, *br*, –N**H_2_**) 5.37–5.137 (dd, –C_6_**H_4_**, J = 6.0 Hz) 2.90 (spt, –C**H**(CH_3_)_2_, J = 6.9 Hz) 2.12 (s, –C**H_3_**) 1.27 (d, –CH(C**H_3_**)**_2_**, J = 6.9 Hz)	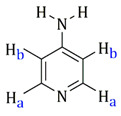

**Table 4 molecules-27-07264-t004:** Cytotoxicity expressed as IC_50_ mean values (mM) of the complexes synthesized and cisplatin exposed to HeLa cells for 48 h.

Compound	IC_50_, mM
Complex **I**	>2
Complex **II**	>2
Complex **III**	>2
Complex **IV**	1.60 ± 0.004
cisplatin	0.06 ± 0.002

**Table 5 molecules-27-07264-t005:** X-ray Data Collection Parameters for **II** and **III**.

	II	III
formula	C_17_H_20_Cl_2_N_2_Ru	C_15_H_20_Cl_2_N_2_Ru
fw	424.32	400.30
cryst color, habit	orange, prism	orange, prism
cryst size (mm)	0.090 × 0.050 × 0.030	0.18 × 0.07 × 0.04
cryst syst	monoclinic	monoclinic
space group	P21/n (#14)	P21/c (#14)
a (Å)	8.9362(9)	16.509(7)
b (Å)	12.8438(12)	13.112(5)
c (Å)	14.9730(16)	7.167(2)
α (°)	90	90
β (°)	91.063(4)	93.90(2)
γ (°)	90	90
V (Å^3^)	1718.2(3)	1547.8(10)
*Z* value	4	4
*D*_calcd_ (g/cm^3^)	1.640	1.718
*F* _000_	856	808
no. of reflns measd	37,748	69,142
no. of observations	5265	4719
no. of variables	208	192
R1	0.0271	0.0180
wR2	0.0563	0.0434
goodness of fit	1.050	0.903

## Data Availability

Data are contained in the article and in Appendix A.

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
