# Peer review of "Synthesis and Characterization of New Ruthenium (II) Complexes of Stoichiometry [Ru(p-Cymene)Cl2L] and Their Cytotoxicity against HeLa-Type Cancer Cells"

_molecules, 2022, doi:10.3390/molecules27217264_

Round 1

Reviewer 1 Report

The manuscript is well written and organized, the results are well explained. The work is useful for researchers in the biological activity of metal complexes field. Some minor revisions should be addressed:

1) In the title, correct ruthenium (ii) to ruthenium (II).

2) The aim of the work should more clarify in the introduction part.

3) The resolution of some figures and structures needs to be improved.

4) The English language needs to be checked very carefully.

Reviewer 2 Report

The manuscript of García et al. describes the synthesis of new four half-sandwich complexes based on the (cymene)ruthenium moiety. The application of these complexes was explored as antitumor agents against HeLa-type cancer cells, where they demonstrated low efficiency. This work is extension of numerous previous investigations of these authors and others. For example, this work is very close to previous work, which are published in Molecules (Pujante-Galián, M.A.; Pérez, S.A.; Montalbán, M.G.; Carissimi, G.; Fuster, M.G.; Víllora, G.; García, G. p-Cymene Complexes of Ruthenium(II) as Antitumor Agents. Molecules 2020, 25, 5063). Although the present paper reports on new complexes, the manuscript is bad written and lacks details and interpretations. So, in the current state, this paper is not of suitable quality for publication in Molecules or is required major revision.   There are several areas of weaknesses that need to be addressed in this paper before publication. 1) The solvent, in which NMR spectra were recorded, should be provided. Moreover, the NMR spectra should be provided in the Supporting Information. 2) On the Figure 4, authors depict intermolecular contacts Ru…Ru. These distances are considerable longer than the sum of the corresponding Van der Waal radii. There are no comments on these contacts in the text. So, they should be deleted from the Figure or discussed in the text. 3) Section 2.3, dealing with the study of cytotoxicity, is poorly written. The data obtained should be compared with any reference compound, for example, cisplatine, as well as related ruthenium complexes (for the same cancer cells, see: Dalton Trans. 2019, 48, 5352; J. Organomet. Chem. 2019, 881, 66). Based on this consideration, relevant general trends and relationships between compound structure and antitumor activity should be proposed. Why did the authors not test the antitumor activity of free ligands (amine and pyridine derivatives) to emphasize the higher activity of the complexes? In addition, Figure 6 should be corrected. The Figure contains diagrams for complexes V-VIII, however these complexes are not presented in the manuscript. In caption to the Figure, there are descriptions for “*” and “***”, but they are not presented in diagrams. 4) I was unable to find the Conclusions section of this manuscript. Thus, the results of this study and their importance remained unclear.

Reviewer 3 Report

Marta G. Fuster et al., explore the study of the Cytotoxicity of New Ruthenium (ii) complexes of

stoichiometry [ru( p-cymene)cl 2l] against  HeLa-type Cancer  3 Cells. The single crystal x-rays diffraction analysis part requires major improvement. So, I am recommending major revision on the basis of following comments.

1.      I check CIF by https://checkcif.iucr.org/. The following ALERT C is found.

PLAT042_ALERT_1_C Calc. and Reported Moiety Formula Strings  Differ     Please Check

Kindly remove it.

2.    In abstract section, kindly replace by words “X-rays diffraction” by “single crystal X-rays diffraction”.

3.    As the crystal structure of complex II is not new. Kindly mention it in abstract section as. The crystal structure of complex III is reported and compared with a close related literature crystal structure II.

4.    Kindly replace Figure 3 by ORTEP diagram so that the thermal vibrations of atoms are visible.

5.    The authors say that in complex II and III, the coordination geometry is octahedron. How much the geometry is deviated from the perfect octahedral geometry?

6.    In Figure 4 and 5, kindly show axes. I think, some infinite chains of molecules are formed by H-bonding. Kindly check.  Only show the H-atoms that are involved in H-bonding. Combine Figure 4 and 5 as one Figure.

7.     The description of supramolecular packing is not fully explored by authors. I highly recommend that the authors should perform Hirshfeld surface analysis of complex II and III as compare the non-covalent interactions. The analysis can be done on Crystal Explorer version 21.5 which is freely available on web for academic use. For guidance, kindly see the following articles 10.1039/B818330A ; 10.1039/b203191b; 10.1080/14756366.2022.2078969

8.    Kindly remove the distances from Figure 4 and 5. Show H-bonding by red dotted lines.

9.    In table 4, kindly correct the symbol of α, β, γ. Correct degree sign.

10.  Kindly mention the name of the Diffractometer on which single crystal XRD data is collected. Reference of the software that is used for absorption correction, crystal structure refinement. How the H-atoms are treated in the refinement. Radiation type used.

11.  As the crystal structure of complex II and III contains pyridine ring. Is there pi-pi stacking interaction in the crystal packing or any other non-covalent interaction that involve pyridine ring?

Round 2

Reviewer 2 Report

The authors have revised the manuscript according to the reviewer's comments, however, some additional minor corrections should be made in this paper before publication:

1)     Please recheck the data of (HR)-ESI-MS spectroscopy in Table 1, because errors are very large compared to calculated values for all compounds.

2)     Please check the melting point for complex IV in Table 1. Compounds with M.p. = 02 are liquids at room temperature.

3)     Please correct the solvent in Table 3. CD2Cl2 should be written instead of CCl2CD2

Author Response

The decomposition point of IV has been changed which was not 2 but 72°. The changes appear in red.
1) eliminating the two values of the mass spectra that were off, 2) changing the IV decomposition point from 02 to 72, 3) changing CD2Cl2 to CD2Cl2. 

Reviewer 3 Report

The authors improve the manuscript in response to all the corrections and suggestions recommended. I am recommending the acceptance of the manuscript in its present form with only a small correction that is the reference 38 is not properly cited. The authors mention that they use crystal Explorer version 21.5 but they provide reference of crystal explorer version 17. Correct and updated reference 38 is 

P.R. Spackman, M.J. Turner, J.J. McKinnon, S.K. Wolff, D.J. Grimwood, D. Jayatilaka, M.A. Spackman, CrystalExplorer: A program for Hirshfeld surface analysis, visualization and quantitative analysis of molecular crystals, J. Appl. Crystallogr. 54(3) (2021) 1006-1011.

I just copy and paste it. Kindly set it according to journal reference style. 

Author Response

According to referee 3, reference 38 has been changed.